



**Effects of sterilization techniques on chemodenitrification and N₂O production in tropical**
**peat soil microcosms**
*Steffen Buessecker[1], Kaitlyn Tylor[1], Joshua Nye[2], Keith E. Holbert[3], Jose D. Urquiza-Muñoz*
*[4,5], Jennifer B. Glass[6], Hilairy E. Hartnett[2,7], Hinsby Cadillo-Quiroz[1,8]\**
[1]School of Life Sciences, Arizona State University, Tempe, Arizona, USA
[2]School Molecular Sciences, Arizona State University, Tempe, Arizona, USA
[3]School of Electrical, Computer and Energy Engineering, Arizona State University, Tempe,
Arizona, USA
[4]Laboratory of Soil Research, Research Institute of Amazonia's Natural Resources, National
University of the Peruvian Amazon, Iquitos, Loreto, Peru.
[5]School of Forestry, National University of the Peruvian Amazon, Pevas 584, Iquitos, Loreto,
Peru
[6]School of Earth and Atmospheric Sciences, Georgia Institute of Technology, Atlanta, Georgia,
USA
[7]School of Earth and Space Exploration, Arizona State University, Tempe, Arizona, USA
[8]Biodesign Institute, Arizona State University, Tempe, Arizona, USA
**Keywords**: soil sterilization, chemodenitrification, abiotic N₂O production, tropical peatlands
**\*Corresponding Author:**
Hinsby Cadillo-Quiroz, LSE-751, 427 East Tyler Mall, Tempe, AZ 85287, Email:
hinsby@asu.edu



## Abstract

Chemodenitrification – the non-enzymatic process of nitrite reduction – may be an important sink for fixed nitrogen in tropical peatlands with low oxygen, low pH, high organic matter, and variable ferrous iron concentrations. Assessing abiotic reaction pathways is difficult because sterilization/inhibition agents can alter the availability of reactants by changing iron speciation and organic matter composition. We compared six commonly used soil sterilization techniques – $\gamma$-irradiation, chloroform, autoclaving, and chemical inhibitors (mercury, zinc, and azide) – for their compatibility with chemodenitrification assays for tropical peatland soils (organic-rich low pH soil from the Eastern Amazon). Out of the six techniques, $\gamma$-irradiation resulted in soil treatments with lowest cell viability and denitrification activity, and least effect on pH, iron speciation, and organic matter composition. Nitrite depletion rates in $\gamma$-irradiated soils were highly similar to untreated/live soils, whereas other sterilization techniques showed deviations. Chemodenitrification was a dominant process in tropical peatland soils assayed in this study. Abiotic $N_2O$ production was low to moderate (3-16% of converted nitrite), and different sterilization techniques lead to significant variations on production rates due to inherent processes or potential artifacts. Our work represents the first methodological basis for testing the abiotic denitrification and $N_2O$ production potential in tropical peatland soil.



## 1 Introduction


Across ecosystems, physical and chemical factors, such as solar radiation or redox gradients, can
drive chemical transformations in the absence of enzymatic catalysis. The nitrogen (N) cycle, in
particular, includes abiotic reactions that can affect the retention of nutrients or substrates (Clark,
1962; McCalley and Sparks, 2009; Parton et al., 2007). Non-enzymatic formation of N-
containing gases has long been known (Jun et al., 1970; Wullstein and Gilmour, 1966). A major
abiotic process in the N cycle is chemodenitrification, the step-wise reduction of nitrite ($NO_2^-$) to
gaseous products, namely nitric oxide (NO), nitrous oxide ($N_2O$) or dinitrogen ($N_2$), often
coupled to iron ($Fe^{2+}$) oxidation, as described in Eq. 1-3 (Davidson et al., 2003; Kampschreur et
al., 2011; Zhu et al., 2013; Zhu-Barker et al., 2015).
$NO_2^- + Fe^{2+} + 2\ H^+ \rightarrow NO + Fe^{3+} + H_2O$ (Equation 1)
$2\ NO + 2\ Fe^{2+} + 2\ H^+ \rightarrow N_2O + 2\ Fe^{3+} + H_2O$ (Equation 2)
$N_2O + 2\ Fe^{2+} + 2\ H^+ \rightarrow N_2 + 2\ Fe^{3+} + H_2O$ (Equation 3)
Eq. 1-2 are plausible in soils and sediments (Jones et al., 2015), however Eq. 3 is likely negligible
in most soil environments because of the unlikely availability of $Cu^{2+}$ at the required
concentrations to reduce $N_2O$ (Moraghan and Buresh, 1977) and relative inertness of $N_2O$.
Anoxic tropical peat soils are expected to have the ideal conditions for chemodenitrification: low-
$O_2$, low pH, high organic matter (OM), and high $Fe^{2+}$ (Kappelmeyer et al., 2003; Nelson and
Bremner, 1969; Porter, 1969; Van Cleemput et al., 1976). In these ecosystems, $NO_x^-$ is supplied
by nitrification fueled by organic N mineralization or from external sources (fertilization, wet or
dry deposition). Abiotic phenol oxidation occurs at oxic-anoxic interfaces in tropical soils, and
may be linked to the N cycle (Hall and Silver, 2013). In such reactions, $NO_2^-$ can be reduced by
phenolic groups to form the nitrosonium cation $NO^+$, which can either (1) remain fixed within the
organic compound as nitrosophenol (Thorn and Mikita, 2000; Thorn et al., 2010), or (2) be



emitted in gaseous form. After tautomerization to an oxime (Raczyńska et al., 2005) and reaction
with $NO^+$ derived from a second $NO_2^-$ ion, hyponitrous acid ($H_2N_2O_2$) can be produced, which
further decomposes to $N_2O$ (Porter 1969; Stevenson et al., 1970) (Eq. 4).

(Equation 4)

Other OM-dependent $NO_2^-$ reduction pathways can produce NO and $N_2$ (McKenney et al., 1990;
Thorn et al., 2010) instead of $N_2O$.

The importance of abiotic N transformations in environmental samples has been

notoriously difficult to quantify due to the artifacts emerging from physical or chemical "killing"
methods intended to eliminate biological activity. In order to distinguish denitrification from
chemodenitrification, enzymes contributing to gaseous N production must be inactivated, most
commonly by addition of sterilants or inhibitors. An efficient sterilization treatment ideally: (1)
contains a negligible number of live cells, (2) eliminates biological activity, and (3) has little or
no effect, directly or indirectly, on abiotic reactions (e.g., it should not alter mineral structure, nor
lyse cells because release of cellular contents could influence abiotic reactions). Because rates
and products of chemodenitrification are dependent on $O_2$, pH, $Fe^{2+}$ concentration and OM
composition, it is important to assess whether a sterilant/inhibitor elicits a physicochemical
change that can affect the availability or interaction of these reactants.

Soil sterilization techniques include γ-irradiation, chloroform ($CHCl_3$) fumigation,

autoclaving, and addition of chemical inhibitors such as mercury (Hg), zinc (Zn), or azide ($N_3$).
Highly energetic γ-irradiation damages enzymes and cell components, rendering cells non-viable
and inactive, generally with minimal effect on soil chemistry (Trevors, 1996). Autoclaving with
high-pressure steam disrupts cell membranes, denatures proteins, and decreases aromaticity and



polycondensation of soil OM (Berns et al., 2008; Jenkinson and Powlson, 1976b; Trevors, 1996).
Fumigation with $CHCl_3$ induces cell lysis and has minimal effect on enzymes (Blankinship et al.,
2014). Chemicals like Hg, Zn, and $N_3$ do the opposite: they inhibit enzymes (Bowler et al., 2006;
McDevitt et al., 2011), but do not lyse cells (Wolf et al., 1989).

We evaluated the appropriateness of six sterilants (γ-irradiation, autoclaving, $CHCl_3$, Hg,

Zn, and $N_3$) for chemodenitrification measurements in low-$O_2$, low-pH, high-OM tropical peat
soils. First, we tested the effects of sterilants on cell membrane viability and biological
denitrification activity. Next, we evaluated the effects of sterilants on soil chemistry (pH, OM
composition, and extractable Fe). Finally, we assessed the effects of the six sterilants on
chemodenitrification measured by $NO_2^-$ depletion and $N_2O$ production.

**2 Materials and Methods**
*2.1 Sample characteristics.* Soil samples were collected in October 2015 from a tropical
peatland, locally known as Quistococha (3°50'S, 73°19'W), near Iquitos (Loreto, Peru). The soil
geochemistry of this site has been described previously (Lawson et al., 2014; Lähteenoja et al.,
2009). The samples were obtained from depths of 15-30 cm below the water table and kept
strictly anoxic during transport and storage at 4°C in the dark. Water saturation and organic
carbon content were determined by oven drying and loss-on-ignition, respectively. Dissolved
organic carbon (DOC) was determined by high-temperature combustion using a Shimadzu TOC-
V Total Organic Carbon Analyzer (Shimadzu Scientific Instruments, Columbia, MD). Inorganic
N species were quantified photometrically using an AQ2 Discrete Analyzer (Seal Analytical,
Southampton, UK) and method EPA-103-A Rev.10 for ammonium ($NH_4^+$; LoD 0.004 mg-N $L^{-1}$,
range 0.02-2.0 mg-N $L^{-1}$) and method EPA-127-A for nitrate ($NO_3^-$) /nitrite ($NO_2^-$; LoD 0.003





mg-N L$^{-1}$, range 0.012-2 mg-N L$^{-1}$). Hydroxylamine was measured photometrically using the
iodate method (Afkhami et al., 2006).
***2.2 Soil sterilization and slurry incubations***. Experiments were started within 6 weeks of soil
collection. For each sterilization procedure, anoxic wet soil was exposed to the chemical sterilant
48 hours prior to start of the NO$_2^-$ incubation or sterilized by physical treatment and allowed to
equilibrate for at least 12 hours. The untreated/live control was incubated as a slurry without any
additions or treatments for 48 hours prior to start of the NO$_2^-$ incubation. Anoxic vials filled with
wet soil were irradiated with a $^{60}$Co source for 7 days, yielding a final radiation dose of 4 Mrad
(40 kGy). The irradiated soil was then prepared for incubation in an anoxic glove box (0.5% H$_2$ in
N$_2$) with disinfected surfaces and sterilized materials to prevent contamination. For autoclaved
samples, soil was prepared for incubation in closed vials and autoclaved at 121°C and 1.1 atm for
90 minutes. The CHCl$_3$-treated samples were fumigated for 48 hours under a 100% N$_2$
atmosphere. Because volatilized CHCl$_3$ corrodes electron capture detectors used for N$_2$O
detection (see below), CHCl$_3$ was removed by flushing the vials with N$_2$ for 5-7 minutes
immediately before the start of incubations.

In contrast to the physical sterilization treatments, soil samples were continuously

exposed to the chemical inhibitors throughout their incubation. Sodium azide (NaN$_3$, Eastman
Organic Chemicals), zinc chloride (ZnCl$_2$, Fisher Scientific) or mercuric chloride (HgCl$_2$, 99.5%,
Acros Organics) were added from anoxic stock solutions to final concentrations of 150, 87.5, and
3.7 mM, respectively. The Hg concentration was the minimum needed to eliminate microbial
heterotrophic growth based on visual inspection of soil extract on agar plates exposed to 0.5 to
92.1 mg L$^{-1}$, which includes concentrations demonstrated to be effective previously (Tuominen et
al., 1994).



After the initial physical or chemical treatment, triplicate incubations were diluted
1:10 in 20 mL of autoclaved 18.2 MΩ·cm water in 60 mL glass serum vials. Triplicate soil
slurries were amended from anoxic, sterile stock solution to a final concentration of 300 μM
$NO_2^-$ (6 μmoles in 20 mL) and sealed with thick butyl rubber stoppers. A parallel set of samples
was amended with 300 μM $NO_3^-$ to evaluate denitrification potential with $CO_2$ measurements.
Control incubations received an equivalent volume of autoclaved 18.2 MΩ·cm water without
$NO_x^-$. Soil microcosms were incubated in the dark at a constant temperature of 25°C. $NO_2^-$ was
quantified in all soil treatments using the Griess assay (Promega, Kit G2930; e.g., Griess 1879).
pH measurements were taken with an Orion 3 Star meter (Thermo Scientific) before and after
sterilization, and at the end of the experiment after 70-76 hours of incubation.
**2.3 Gas chromatography.** To quantify $N_2O$ and $CO_2$ production, 200 μL of headspace gas was
sampled with a gas-tight syringe (VICI Precision Sampling) and injected onto a gas
chromatograph (GC, SRI Instruments) equipped with both an electron-capture detector (ECD)
and a flame-ionization detector (FID). Two continuous HayeSep-D columns were kept at 90°C
(oven temperature); $N_2$ (UHP grade 99.999%, Praxair Inc.) was used as carrier gas, and $H_2$ for
FID combustion was supplied by a $H_2$ generator (GCGS-7890, Parker Balston). For $CO_2$
measurements, a methanizer at 355°C was run in line before the FID. The ECD current was 250
mV and the ECD cell was kept at 350°C. The $N_2O$ and $CO_2$ measurements were calibrated using
customized standard mixtures (Scott Specialty Gases, accuracy ±5%) over a range of 1-400 ppmv
and 5-5,000 ppmv, respectively. Gas accumulation in the incubation vials was monitored over
time. Gas concentrations were corrected using Henry's law and the dimensionless concentration
constants $k_H^{cc}(N_2O) = 0.6112$ and $k_H^{cc}(CO_2) = 0.8313$ (Stumm and Morgan, 2012) to account for
gas partitioning into the aqueous phase at 25°C.





**_2.4 Live/dead cell staining._** To assess the efficacy of sterilants or inhibitors visually, the bacterial
viability kit LIVE/DEAD BacLight L7012 (Molecular Probes, Invitrogen) containing SYTO9
and propidium iodide dyes was used to stain and distinguish dead and living cells on the basis of
intact cell walls. The green (live) and red (dead) signals were counted at 60x magnification from
10 squares of 0.01 mm$^2$ randomly distributed in the center of a 5 µL Neubauer chamber, using an
Olympus BX-61 microscope with the FITC/Cy5 filter set. Photographs were taken with an
Olympus DP-70 camera attached to the microscope. Particles were counted with ImageJ software
version 1.50i (Abràmoff et al., 2004).
**_2.5 Fe extraction and quantification._** Dissolved Fe species were extracted from peat soil
incubations following the protocol of (Veverica et al., 2016). The method is based on an ionic
liquid extraction using _bis_-2-ethylhexyl phosphoric acid (Pepper et al., 2010), which was shown
to be more suitable for extraction of Fe from humic-rich matrices than the traditional ferrozine or
phenanthroline methods. Briefly, 2.5 mL of soil slurry was filtered (0.2 µm nylon filter; Celltreat
Scientific Products) and mixed with 7.5 mL of HCl (0.67 N) in an extraction vial in an $N_2$ glove
box. The $O_2$ concentration in the glove box was continuously monitored and remained <10 ppm.
To separate $Fe^{3+}$ from $Fe^{2+}$, 10 mL of 0.1 M _bis_-2-ethylhexyl phosphate (95%, Alfa Aesar) in _n_-
heptane (99.5%, Acros Organics) was added to the acidified sample. Next, the organo-aqueous
emulsion was shaken at 250 rpm in closed extraction vials for 2 hours. The _bis_-2-ethylhexyl
phosphate chelates $Fe^{3+}$ more effectively than it chelates $Fe^{2+}$. The $Fe^{2+}$-containing aqueous phase
was sampled into a 3-fold HCl-washed HDPE vial (Nalgene) in the glove box. The $Fe^{3+}$ fraction
chelated in the organic phase was then back-extracted into an aqueous phase by the addition of 10
mL 4N HCl and shaking at 250 rpm in closed extraction vials for 20 minutes. $Fe^{3+}$ and $Fe^{2+}$
fractions were quantified separately in acidified aqueous solution by inductively coupled plasma-
optical emission spectrometry (ICP-OES; Thermo iCAP6300 at the Goldwater Environmental





Laboratory at Arizona State University). The ICP-OES pump rate for the Ar carrier was set to 50
rpm and Fe2395 and Fe2599 lines were used for Fe quantification. Iron concentrations were
determined from a calibration curve (0.01-10 mg L$^{-1}$) by diluting a standard solution (100 mg L$^{-1}$,
VHG Labs, product # SM75B-500) in 0.02 N $HNO_3$.
***2.6 Dissolved organic matter fluorescence analysis.*** 3D-fluorescence analysis was performed on
a Horiba Jobin-Yvon Fluoromax 4 spectrofluorometer. Excitation-emission matrices (EEMs)
were generated by obtaining emission spectra ($\lambda_{Em}$ = 300-550 nm, at a step size of 2 nm) at
excitation wavelengths from 240-450 nm at a 10 nm step size. All EEMs were blank corrected
and normalized daily to the Raman peak of ultrapure water (deionized, carbon-free,18.2 MΩ·cm;
Barnstead$^{tm}$ NanoPure). The samples were taken at the same time as those for Fe analysis. Prior
to analysis, soil slurries were filtered using a solvent-rinsed Whatman GF/F filter (nominal pore
size 0.7 μm) to obtain ~10 mL filtrate. Samples were diluted with ultrapure water if their UV
absorbance exceeded 0.3 so that inner-filter corrections could be made (Stedmon, 2003). We
calculated total fluorescence as the matrix sum of all signals in the EEM. Fluorescence indices
were used to characterize various classes of fluorophores in the dissolved organic matter (DOM)
pool. Fluorescence Index (FI) was calculated as the sum of the intensity signal in the emission
spectra from 470-520 nm collected at an excitation wavelength of 370 nm (Cory and McKnight,
2005). Humification index (HIX) was determined from the peak area under the emission
spectrum from 435–480 nm divided by the area from 300–445 nm, both collected at an excitation
wavelength of 254 nm (Ohno, 2002). The "freshness" was determined as β/α, the ratio of
emission intensity at 380 nm to the emission intensity maximum between 420 and 435 nm, both
collected at an excitation wavelength of 310 nm (Wilson and Xenopoulos, 2009).
***2.7 Statistical Analyses.*** All basic statistical tests were performed with JMP Pro software
(Version 13.1.0, SAS Institute Inc., Cary, NC, USA).




**3 Results**
*3.1 Composition of high-OM tropical soils.* The tropical peat soil used for the incubation
experiments had 5.5-5.8 pH, 92.2% water content, 307±5 mg TOC $g^{-1}$ dry weight, and 3.8±0.9 g
total Fe $kg^{-1}$ soil. The extractable iron fraction partitioned as 54±3 μM extractable $Fe^{3+}$ and
213±16 μM extractable $Fe^{2+}$. The native soil pore water had 13.2±1.2 mg $L^{-1}$ DOC, 436±79 μg N
$L^{-1}$ $NH_4^+$, 9.7±1.3 μg N $L^{-1}$ $NO_3^-$, and 3.9±0.2 μg N $L^{-1}$ $NO_2^-$. Hydroxylamine was below
detection in all cases (<3 μM). Soil pH dropped from 5.5-5.8 in untreated soil to 3.6, 4.8, 5.0, 5.2,
and 5.4 after treatment with Hg, Zn, γ-irradiation, autoclaving, and $CHCl_3$, respectively. Only $N_3$
treatment increased soil pH (to 6.4).

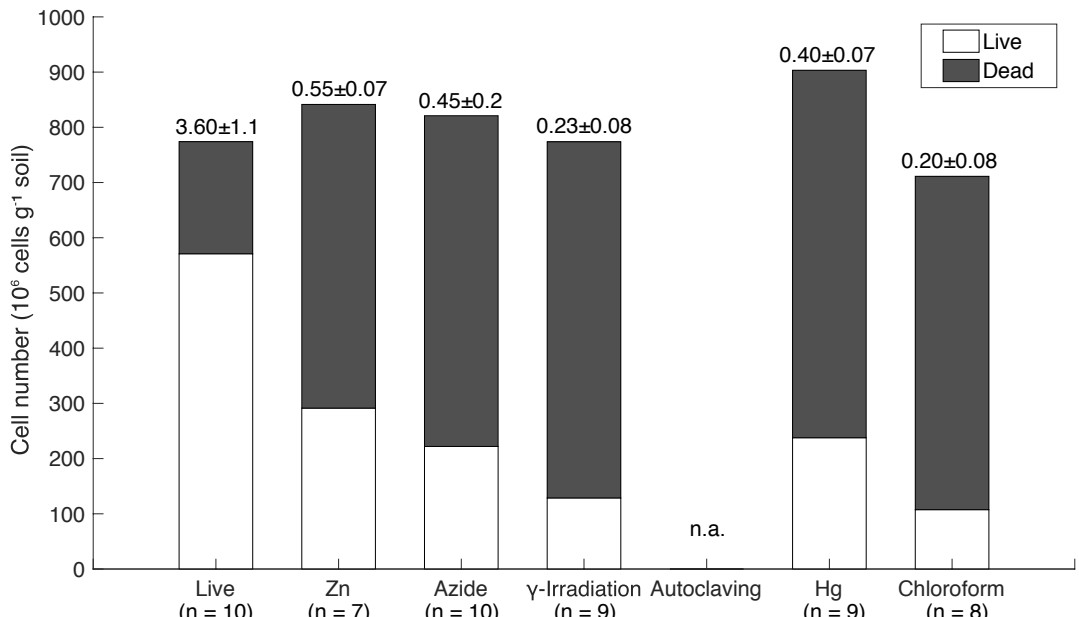

**Figure 1. Live/dead microbial cell counts of tropical peatland soils**. The numbers above the
bars indicate the live to dead signal ratio ± SD. No detectable signal was observed in autoclaved
samples.




*3.2 Effects of sterilants on cell integrity and potential of denitrifying activity.* Live/dead dyes
were used to assess microbial viability by means of membrane integrity, where a "dead" signal
indicates disrupted or broken cell membranes (Stiefel et al., 2015). The majority (74%) of cells in
the live incubation displayed the "live" signal (**Fig. 1**). The $CHCl_3$ and γ-irradiated treatments
were most effective at reducing the number of viable cells (~15% intact membranes after
sterilization). Chemical inhibitors (Hg, Zn, and $N_3$) were less effective at killing cells (~30%
intact membranes after sterilization). Autoclaved samples did not fluoresce, likely due to cell
lysis during steam pressurization.


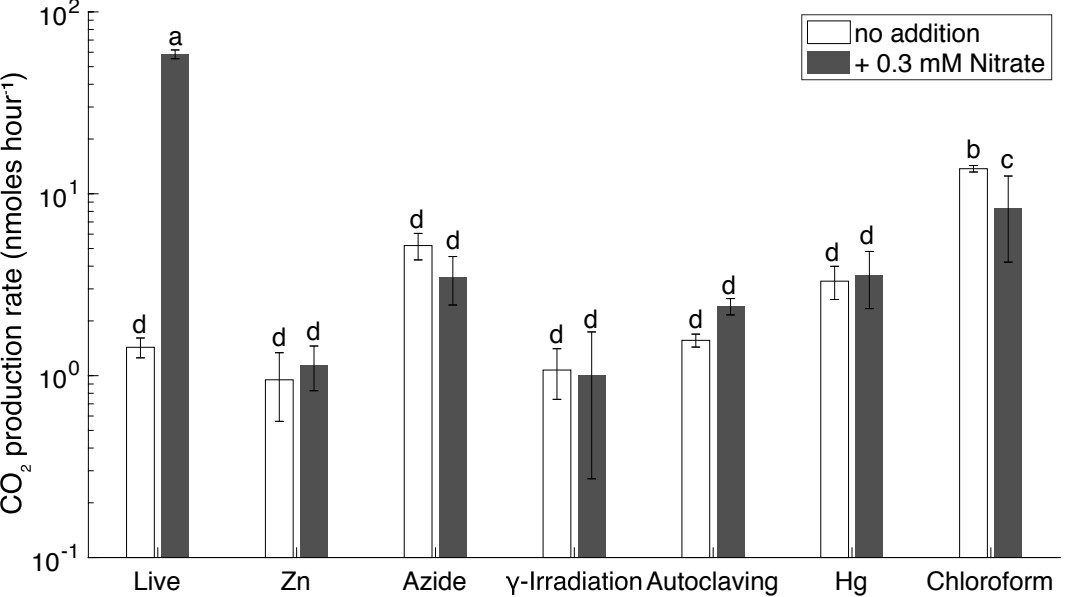


**Figure 2. $CO_2$ production rates in 3-day soil slurry incubations of Quistococha peat soil**
**amended with and without 0.3 mM $NO_3^-$.** Error bars are one SD (n=3). Columns marked with
the same letter are not statistically different from each other (Student's *t*, $p > 0.05$, n=3).




Biological denitrification activity was measured over three days in live and sterilized soils
based on the difference in $CO_2$ production with and without added $NO_3^-$. An efficient sterilization
treatment would show no changes in $CO_2$ beyond that due to equilibration between the gas phase
and aqueous phase. Nitrate stimulated $CO_2$ production in live soil (ANOVA, $p < 0.05$) and not in
the γ-irradiated, Zn, Hg, N₃, or autoclaved incubations **(Fig. 2)**, indicating that residual cells in
the sterilized treatments were not capable of denitrification.

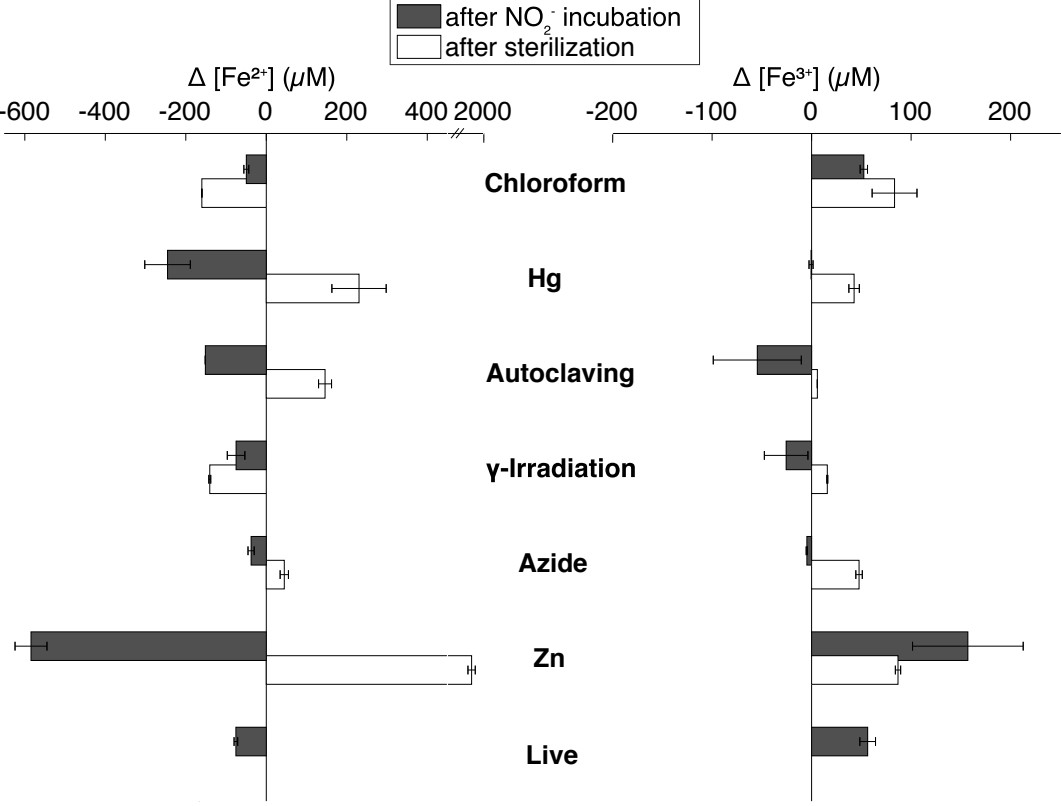



**Figure 3. Changes in extractable $Fe^{2+}$ (left) and $Fe^{3+}$ (right) concentration in Quistococha**
**peat soil incubations after sterilization (difference between sterilization baseline and live**





**baseline value) and after $NO_2^-$ amendment and incubation (difference between $NO_2^-$ and**
**control incubations).** Note the difference in scales. Values represent the extractable fraction of
both species. Error bars are one SD (n=2).

*3.3 Effects of sterilants on soil chemistry*. In general, sterilization increased extractable $Fe^{2+}$ and
$Fe^{3+}$ relative to live controls (**Fig. 3**). This trend was particularly pronounced in Zn treatments,
which had 9x higher extractable $Fe^{2+}$ (1915±26 μM) and 1.6x higher extractable $Fe^{3+}$ (87±3 μM)
than live controls. The Hg treatment showed the second largest increases. In the presence of
$NO_2^-$, extractable $Fe^{2+}$ decreased and extractable $Fe^{3+}$ increased in live, Zn, and $CHCl_3$-fumigated
treatments, as expected if $Fe^{2+}$ was oxidized by $NO_2^-$ during chemodenitrification. However,
autoclaving, γ-irradiation, and $N_3$ lowered $Fe^{3+}$ concentrations, suggesting the influence of
unknown concomitant reactions. For instance, autoclaving (largest drop in $Fe^{3+}$) already showed
lower $Fe^{3+}$ concentrations after sterilization. Production of $Fe^{3+}$-reduction artifacts in treatments
could lead to $Fe^{3+}$ depletion and, hence, mask increase in $Fe^{3+}$ due to chemodenitrification. $NO_2^-$
addition resulted in near-complete depletion of extractable $Fe^{2+}$ in live, $CHCl_3$-fumigated, and γ-
irradiated soils. Changes in Fe speciation with other sterilants were more moderate. Minimal
changes were observed for other metals (e.g., Mn, Al, Cu, and Zn; data not shown).











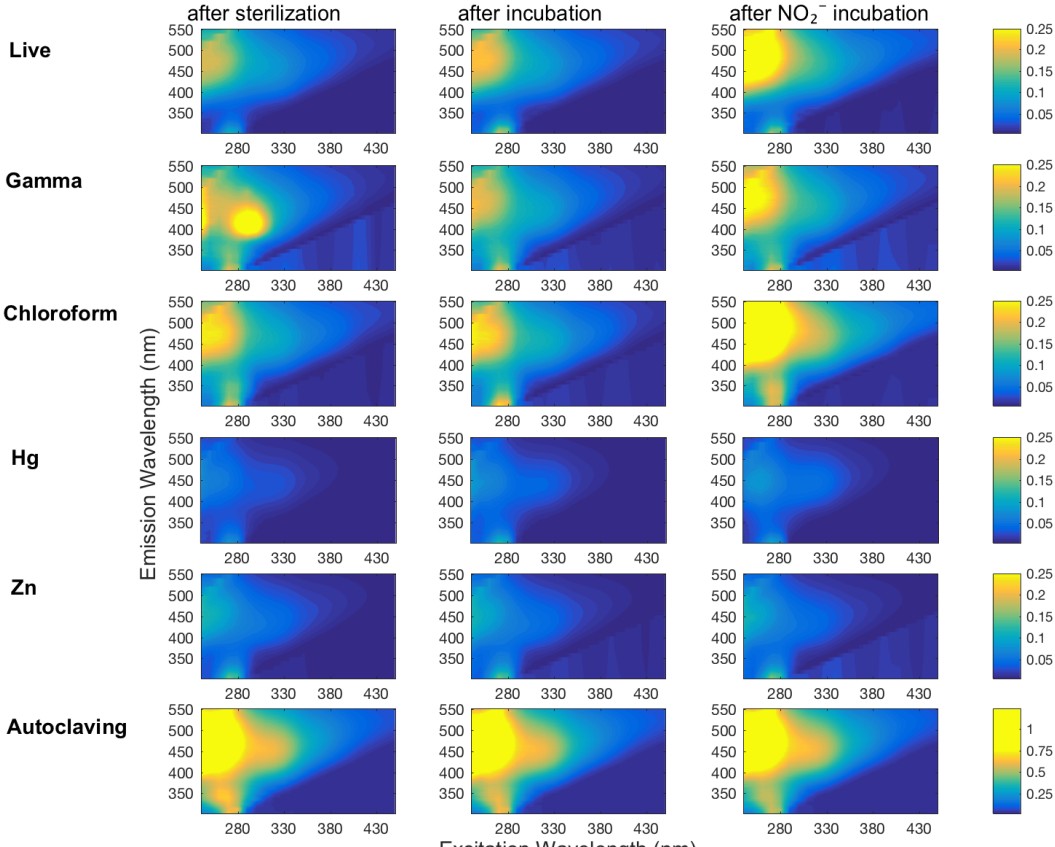

**Figure 4. Representative plots of DOM fluorescence in soil slurry incubations of**

**Quistococha peat soils.** DOM fluorescence is presented as excitation-emission matrices (EEMs)

collected for each treatment (rows) after the sterilization procedure or live control (left column),

after incubation with no amendment ("after incubation" control, middle column), and after

incubation with 300 μM $NO_2^-$ (same time point as control, right column). The colored bar shows

the individual signal intensity. All but "autoclaving" treatment has same scale of signal intensity,

autoclaving effects increased about 5 times the signal intensity scale.





Fluorescence analysis of soil extracts using excitation-emission matrices (EEMs) was
used to evaluate changes in DOM containing aromatic moieties or conjugated double bonds
(Stedmon et al., 2003); **Fig. 4**). The $N_3$ treatment was excluded from this analysis due to an
interference with $N_3$ absorbance that prevented inner-filter corrections from being made. The
EEM signals showed the greatest change in the "humic" region ($\lambda_{Ex}$ <240-270 nm, and $\lambda_{Em}$ =
460-500 nm; (Fellman et al., 2010), especially in Zn and Hg treatments, which significantly
increased the FI to 1.49 (**Table 1**). Zn and Hg may elicit direct fluorescence quenching by the
formation of Zn and Hg metal complexes (McKnight et al., 2001) or possibly due to indirect
quenching by higher dissolved $Fe^{2+}$. Signal strength in the humic region was enhanced by $NO_2^-$
addition in the live, $CHCl_3$-fumigated, and γ-irradiated treatments. All five sterilization
treatments had lower aromaticity (HIX) than live controls (**Table 1**). Autoclaved samples had
tenfold higher total fluorescence compared to live soils, suggesting that autoclaving degraded
insoluble humics into more soluble and less condensed OM.
***3.4 Effects of sterilants on chemodenitrification and abiotic $N_2O$ production***. In the first 48
hours, $NO_2^-$ consumption rates were the highest in live soil (5.2 μM $h^{-1}$), closely followed by
irradiated samples (4.5 μM $h^{-1}$, **Fig. 5**). The major chemodenitrification pathway for $N_2O$
formation was likely $NO_2^-$ reduction by $Fe^{2+}$, resulting in consumption of ~1.5 μmol $Fe^{2+}$ and
accumulation of ~1.1 μmol $Fe^{3+}$ in the live control (**Fig. 3**). After 48 hours, $NO_2^-$ depletion
continued to completion in the live control but slowed in all treatments other than the metal
additions. After 72 hours of incubation, 3-16% of $NO_2^-$-N was converted to $N_2O$-N across
treatments. Higher $N_2O$ production rates were observed in live, $Zn^{2+}$, and $N_3^-$ treatments (0.5-0.7
nmol $N_2O$ $g^{-1}$ $h^{-1}$, $r^2$ > 0.95) than in γ-irradiated, $CHCl_3$-fumigated, autoclaved, and Hg treatments
(0.1-0.2 nmol $N_2O$ $g^{-1}$ $h^{-1}$, $r^2$ > 0.9). Production rates within treatments showing high or low rates
were not significantly different (Student's $t$, $p$ >0.05) although comparisons across treatments



with high or low rates were statistically different (Student's $t$, $p < 0.05$). Thus, we identified a
higher and lower group of sterilant-dependent $N_2O$ production rates from the same soil samples.
The live control showed logarithmic $N_2O$ accumulation while the sterilized treatments had linear
accumulation over time, the later as expectable in abiotic accumulation (**Fig. 5**).

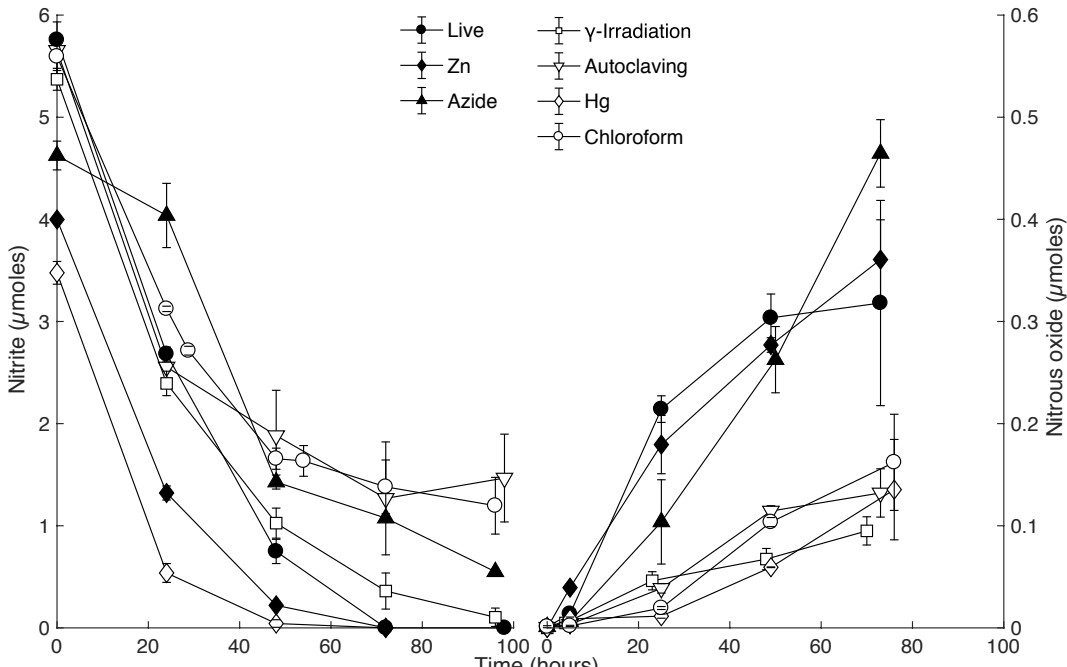



**Figure 5. $NO_2^-$ consumption (left) and $N_2O$ production (right) for different sterilant**
**treatments in soil slurry incubations of Quistococha peat soil.** Both N species were
simultaneously measured in all treatments. The product yield represents $N_2O$-N as molar fraction
of $NO_2^-$-N. Note the difference in left and right y-axis scales. Error bars are one SD (n=3).





## 4 Discussion

### 4.1 Chemodenitrification is a dominant $NO_2^-$ consumption process in slurry incubations of tropical peat soils.
Similar $NO_2^-$ consumption rates between live and irradiated treatments imply that $NO_2^-$ depletion was dominated by abiotic processes over the first 48 hours. In general, abiotic reactions tend to be linear processes, whereas microbially mediated reactions can be affected by enhanced expression of genes or cell reproduction in a nonlinear fashion (Duggleby, 1995). The difference in linearity of $N_2O$ production in sterilized vs. live treatments (**Fig. 5**) suggests that biological denitrification did not occur in sterilized soils.

Compared to our study, incubations of artificial media with 200 μM $NO_2^-$, 0.5-8.1 mM $Fe^{2+}$, and a pH of 7-8 had similar rates of $Fe^{2+}$ depletion but 10x higher rates of $NO_2^-$ reduction, and higher (~10-50%) $N_2O$ yields (Buchwald et al., 2016; Jones et al., 2015). In our peat incubations, reactive OM likely trapped $NO_2^-$ in the soil matrix via OM-bound nitrosation reactions (Thorn and Mikita, 2000; Thorn et al., 2010) and the lower pH likely promoted conversion of $NO_2^-$ to NO (Kappelmeyer et al., 2003; Porter, 1969) or $N_2$ (Stevenson et al., 1970). Studies in low pH northern temperate peat soils, have shown the primary product of abiotic $NO_2^-$ reduction was NO, not $N_2O$ (McKenney et al. 1990).

### 4.2 Artifacts due to sterilization methods for chemodenitrification assays.
Azide and Zn exhibited enhanced $NO_2^-$ conversion to $N_2O$, at rates at least twice to five times as high as those measured for the other sterilants (**Fig 5**), likely due to higher pH and Fe availability, respectively. In the $N_3$ treatments, elevated $N_2O$ production could be explained by the reaction of protonated $NO_2^-$ with $N_3$ in a pH dependent manner (Stedman, 1959), plus other changes in soil solution originated from the increase of pH. Nitrite reaction with $N_3$ has been characterized in marine and freshwater solutions reaching its maximum at pH 4.5 and proceeding slowly yet significantly (20% conversion in 1 hour) at pH > 5 (McIlvin and Altabet, 2005) as in our slurries. Moreover,



$N_3$'s self-fluorescence impeded OM measurements, making $N_3$ an incompatible sterilizing agent
for chemodenitrification studies. Zn increased Fe availability and may have increased $NO_2^-$
affinity for reactive OM groups; both effects would lead to an abiotic increase in $N_2O$ production
(Clark, 1962; McCalley and Sparks, 2009; Parton et al., 2007). Zinc treatment lowered the soil
pH, which may have promoted cation displacement and stability of dissolved $Fe^{2+}$ (Hutchins et
al., 2007), thus enhancing $N_2O$ production. Several studies have used Zn treatments as valuable
agent for field applications (Babbin et al., 2015; Ostrom et al., 2016). Zn is less hazardous to
humans than some of the other sterilants. We propose that the use of Zn could provide useful
information about abiotic *in-situ* rates as long as Zn-induced chemodenitrification is accounted
for. A correction could be applied if a complementary laboratory assessment (using the more
efficient γ-irradiation) were used to develop an ecosystem-specific correction factor.

Divalent $Hg^{2+}$ can be abiotically methylated by fulvic acid-type substances (Rogers,

1977). The reaction oxidizes OM and can diminish its reducing power as indicated by decreased
reactivity of humic acid with $NO_2^-$ (Gu et al., 2011; Zheng et al., 2011) thus interfering with the
abiotic assay. Another potential factor associated with the Hg treatments is metal sorption. At
low pH (3.6), 98% of Hg was sorbed to humic acids, whereas only 29% of Zn was sorbed at pH
~4.8 (Kerndorff and Schnitzer, 1980). Full sorption capacity of peat is presumably reached in
seconds (Bunzl et al., 1976) and the differing sorption behavior of Hg and Zn may play a role in
the reaction potential of $NO_2^-$ with OM. It has been demonstrated that Hg introduced into peat
soil leads to sorption of Hg ions to various functional groups, including phenols (Drexel et al.,
2002; Xia et al., 1998). Hence it is plausible that Hg sorbed to functional groups subject to
electrophilic attack by $NO^+$(e.g., nitrosophenol, Eq. 3) may hamper nitrosation, and therefore
protect OM from reacting with $NO_2^-$. This could lead to a selective suppression of the OM-
dependent $N_2O$ production pathway.





Chloroform fumigation resulted in potential $N_2O$ production rates within the lower

production range treatments with minor differences in Fe speciation and DOM fluorescence.
However, unlike the other sterilized samples, $CHCl_3$-fumigated samples showed enhanced $CO_2$
production stimulated by $NO_3^-$ addition. Removal of $CHCl_3$ from our samples before substrate
addition could have provided an opportunity for a few surviving heterotrophs to re-grow and use
the easily-degradable organic material derived from dead cells. Indeed, chloroform can lyse cells,
providing substrates for growth to $CHCl_3$-resistant microorganisms (Zelles et al., 1997).
Continued exoenzyme activity has been also described as a $CO_2$ source: however, this would not
include denitrification enzymes, since none enzymes involved in the denitrification pathway are
exoenzymes (Blankinship et al., 2014; Jenkinson and Powlson, 1976a). Chlorination of natural
OM may prompt formation of quinones (Criquet et al., 2015), which are intermediates in the OM-
based abiotic $N_2O$ production (Thorn and Mikita, 2000); indeed, regions of the EEMs
corresponding to hydroquinones (Cory and McKnight, 2005) appear to be slightly higher in
$CHCl_3$ treatments. The benzene derivative produced during nitrosophenol reaction with $NO_2^-$
leads to reduced π-electron delocalization (Eq. 4). Because excitation of π-electrons produces
fluorescence, reactions with $NO_2^-$ might be expected to reduce OM fluorescence. However, the
experiment duration is important and if indeed microbial cells reproduce after the treatment, short
experimental periods (e.g., hours or days) or reapplication of $CHCl_3$ might keep down the
numbers of any potential denitrifiers improving the use of this method.

Autoclaved peat soil revealed abiotic $N_2O$ production rates close to the average of the

lower production range group, accompanied by but ICP-OES and fluorescence spectroscopy
results also showed significant changes in Fe speciation and DOM composition. EEMs
demonstrate lower values for the HIX in autoclaved peats (**Table 1**), consistent with fluorescence
data from a study that demonstrated a decrease in the aromaticity and polycondensation of soil



extracts from autoclaved soil (Berns et al., 2008). Autoclaving likely caused degradation and
solubilization of insoluble humic components. The direct effects of autoclaving are very much
dependent on the heat and pressure stability of the indigenous soil constituents, but the substantial
soil structural changes likely introduce chemical artifacts that are absent in the native live soil.
***4.3 Gamma irradiation is the preferred sterilization method for chemodenitrification assays.***
The fewest chemical artifacts were observed in γ-irradiated samples. Soil that had been exposed
to γ-rays showed the lowest $N_2O$ production rates, approximately one-fifth of those observed in
live samples. Irradiation also caused only very small changes in Fe speciation relative to live
controls and yielded EEMs that were remarkably similar to those obtained from live soil extracts.
Our measurements of sterility and respiratory activity indicated the lowest potential for biological
activity and hence, the least amount of interference for the time period tested. We therefore
confirmed γ-irradiation to be a preferred method for sterilizing soil (Trevors, 1996) and for
assessing abiotic $N_2O$ production potentials. In practice, the long preparation time needed to
reach a sufficient dose (dependent on radiation source, see **Methods**) was compensated for by the
lack of chemical artifacts during the experiment and the reduced number of hazardous waste
products. Limited accessibility to irradiation facilities and the absence of a field portable option
remain the main challenges to wide distribution of this approach.









**Table 1. Characteristics of dissolved organic matter in soil extracts from incubations of peat from Quistococha, Peru.** FI, HIX, and freshness indices were calculated as described in the methods section. The "tyrosine-like" region is defined at an excitation of 270-275 nm and an emission of 304-312 nm (Fellman et al., 2010). The signal for that region was averaged across replicates and expressed as percent difference between $NO_2^-$ additions and controls ± standard deviation of replicates. A drop in the signal intensity was consistently apparent, clear differences between the treatments were not, due to high standard deviation of replicates.

| Treatment | | FI** | | HIX*** | | Freshness | Drop in mean fluorescence of the "Tyrosine-like" region (% over control) |
|---|---|---|---|---|---|---|---|
| **Live soil** | *Baseline* | 1.20 | *a* | 5.57 | *a* | 0.44 | |
| | *Control* | 1.21 | | 4.72 | | 0.41 | |
| | *Nitrite added* | 1.16 | * | 7.11 | * | 0.40 | 12.1±6.1 |
| **Zn** | *Baseline* | 1.49 | *b* | 2.70 | *b* | 0.58 | |
| | *Control* | 1.50 | | 2.27 | | 0.59 | |
| | *Nitrite added* | 1.55 | * | 2.05 | | 0.62 | 5.9±4.0 |
| **Autoclaving** | *Baseline* | 1.20 | *a* | 2.54 | *b* | 0.47 | |
| | *Control* | 1.20 | | 2.83 | | 0.46 | |
| | *Nitrite added* | 1.20 | | 2.97 | | 0.43 | 31.5±24.6 |
| **Chloroform** | *Baseline* | 1.23 | *c* | 2.79 | *b* | 0.43 | |
| | *Control* | 1.27 | | 2.70 | | 0.44 | |
| | *Nitrite added* | 1.14 | * | 4.12 | * | 0.40 | 13.5±6.4 |
| **γ-Irradiation** | *Baseline* | 1.30 | *d* | 1.90 | *b* | 0.57 | |
| | *Control* | 1.27 | | 2.35 | | 0.56 | |
| | *Nitrite added* | 1.21 | * | 2.95 | | 0.52 | 2.4±0.8 |
| **Hg** | *Baseline* | 1.49 | *b* | 2.20 | *b* | 0.57 | |
| | *Control* | 1.50 | | 1.60 | | 0.56 | |
| | *Nitrite added* | 1.44 | * | 2.12 | | 0.51 | 13.8±3.9 |

\* indicates significant difference to control.
\*\* Fluorescence index.
\*\*\* Humification index.
Mean values marked with the same letter are insignificantly different from each other.

427



*5 Conclusion*

High $N_2O$ emissions occurs in tropical regions with water-saturated soils (Liengaard et al., 2014;

Park et al., 2011; Pérez et al., 2001). Whether these tropical N emissions are solely biotic or have

abiotic contributions is not well known, because rates of chemodenitrification are not commonly

evaluated. Abiotic processes in the N cycle remain overlooked, partly due to the lack of reliable

means of quantifying abiotic reactions. This study showed that chemodenitrification occurs in a

tropical peat soil, leading to a low to moderate fraction of $N_2O$ conversion from nitrite

amendment. We also demonstrated that γ-irradiation is the "gold standard" for

chemodenitrification assays. The application of $N_3$ to quantify abiotic $N_2O$ production is

unsuitable because changes associated to fraction of the sterilant itself may react to form $N_2O$ and

effects increased pH. $CHCl_3$ and γ-rays have slightly reducing effects on the soil Fe pool and

might lead to a weak discrimination against pathways involving Fe as reactant. $CHCl_3$ fumigation

was another approach with limited effects on Fe chemistry that lowered the number of viable

cells greatly, however, the potential for microbial regrowth after $CHCl_3$ removal is its main

drawback. Autoclaving seemed to have minor disadvantages on abiotic $N_2O$ production, despite

the substantial changes to soil OM.

Unlike other lab-intensive treatments, the application of Zn and Hg are amenable for field

experiments; however, we observed distinct chemical artifacts when using both of these options.

Care is warranted if using Zn and Hg chemical inhibitors, which can increase Fe availability and

may thus overestimate Fe-dependent abiotic $N_2O$ production rate. A potential disadvantage of the

application of toxic metals is a decrease in soil pH. We cannot exclude pH-driven effects on N

intermediates; however, no major deviation in the final $N_2O$ production rate related to

acidification was observed. With the methodological evaluation presented here, we determined

that a directed selection of approaches can allow for better constrained and more detailed studies



of the role of abiotic pathways and soil components shaping denitrification and $N_2O$ fluxes from
soil ecosystems.

**Acknowledgements**
We thank Chris Laurel, Roy Erickson, and Cathy Kochert for training and assistance with the
ICP-OES analysis at ASU's Goldwater Environmental Laboratory, and Steven Hart for advice
optimizing the epifluorescence microscopy. We also thank Nabil Fidai, Jaime Lopez, Analissa
Sarno and Mark Reynolds of the Cadillo Lab for their enduring support during the experimental
phase. This work was funded by an NSF-DEB award (#1355066) to H.C-Q and a NASA award
(NNX15AD53G) to H.E.H and H.C-Q. The results reported herein also benefited from
collaborations and/or information exchange within NASA's Nexus for Exoplanet System Science
(NExSS) research coordination network sponsored by NASA's Science Mission Directorate. All
data presented in this paper is available in the Dryad Digital Repository.

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





**Competing Interests Statement**
The authors have no competing interests to declare.