# Peer review of "Effects of sterilization techniques on chemodenitrification and N₂O production in tropical peat soil microcosms"

_Biogeosciences, 2019_

## Referee Comment (RC1) · Anonymous Referee #1 · 13 Aug 2019

The article, Effects of sterilization techniques on chemodenitrification and N2O production in tropical peat soil microcosms, is well written and details a useful methodological study. Abiotic N2O production processes and links between N and Fe cycling are often overlooked but potentially important processes in some environments. More clear and consistent methodologies to measure such processes are needed and this article does a good job outlining potential limitations of different sterilization methods and the use of EEMs to characterize changes in organic matter was a novel addition. The article is suitable for publishing with minor edits. L23-25: suggest splitting into two sentences and making the second sentence a more concrete statement of the reactants/conditions necessary for chemodenitrification, similar to that of L84-87.

Not all readers readily familiar with chemodenitrification. L34: Consider adding, "of NO2- consumption" to the sentence, "dominant process..." L34-35: Consider defining abiotic N2O production as one endpoint of chemodenitrification. L50-52: Example of use of 'non-enzymatic' and 'abiotic'. Please define and then be consistent with use of non-enzymatic vs abiotic. Suggest using just one term, abiotic is more common, I believe. L56-58: Check equations. Are H2O and H+ flipped? L60: Cu2+? Should this be Fe2+? L66-L72: Found this section a bit confusing. A sentence or two introducing and/or contextualizing these reactions would help readers who are not familiar with these processes. Suggest adding an explanation of how these processes could be affected by sterilization techniques, if indeed that is the point of this part of intro. . . L73-74: Equations absent L124: Were all treatments prepared in an anaerobic glove box prior to incubation? L175: N2 or H2 glove box? Not sure if it would matter, just checking for consistency L268: Zn data is shown in Fig. 3, referring to another figure or a typo? L292: increased the FI relative to what? Suggest reminding readers of what the baseline and controls were for Table 1, or maybe I missed this explanation earlier. Not sure if referring to a change in time or change relative to live or relative to +/- nitrite etc. L307: where is r2 value coming from, was there a regression analysis done? L327: However, to me many of the NO2- consumption lines do not look highly linear in first 48 hrs (Fig. 5). L380: none -> no L392: delete 'accompanied by'? or check grammar of this sentence

---

## Referee Comment (RC2) · Anonymous Referee #2 · 6 Sep 2019

Review of "Effects of sterilization techniques on chemodenitrification and N2O production in tropical peat soil microcosms" by Buessecker et al.

Summary

Understanding the relative contribution of chemodenitrification is severely hampered by artifacts that may arise as an unintended result of such sterilization methods. The authors present a very thorough assessment of the impact of commonly used sterilization techniques on determination of abiotic nitrogen reaction mechanisms in soils. Specifically, they have evaluated addition of mercuric chloride, zinc chloride, sodium azide and chloroform as biocide agents – as well as treatment by autoclaving and gamma irradi-

ation. Abiotic reactions involving nitrogen intermediates (NO2-, NO, etc.) are known to commonly involve redox active metals (Fe, Mn) and organic matter complexes. As presented in the manuscript, the authors' approach focused on a comprehensive evaluation of the impact of sterilization methods on, bulk soil properties (pH), availability and speciation of metals (Fe), resultant impact on microbial cell status (live/dead), the composition of dissolved organic matter, the consumption rate of amended NO2- and the production rates of CO2 (as a proxy for heterotrophy) and N2O. This study is an exceptionally comprehensive assessment of the suitability of these sterilization techniques in the context of soil environments – with clear implications for aquatic systems as well. The authors find that gamma-irradiation results in the least severe generation of artifacts – and therefore offers the clearest path towards realistically constraining abiotic reaction mechanisms (such as chemodenitrification) in environmental studies.

The manuscript is very well written and organized, the results are presented in clear and concise figures and the discussion provides succinct and compelling arguments for explaining the observations. While there are a million different permutations of reaction conditions that could/should be evaluated – in my opinion – this study represents a very nice benchmark for the study of abiotic nitrogen reactions for future studies and warrants publication in Biogeosciences.

Minor Comments

L 60: explain that Cu is required for nitrous oxide reductase. L 73: Equation 4 is really more of a Figure, no? L155: Explain what a methanizer is. It may be pertinent to mention that decomposition of nitrous acid (HNO2) could be occurring at the especially low pH conditions of the HgCl2 addition. Might be useful to consider whether you think this could be contributing to any of the dynamics observed in this treatment, where pH was 3.6. See Park and Lee, 1988 (J. Phys. Chem. v92. p6294). L380: . . . since no enzymes. . . L392: . . . accompanied by ICP-OES. . .

---

## Author Comment (AC1) · 4 Oct 2019

We thank the reviewer for positive comments and pointing to a few improvement points. Below find the specific answers to each one

L23-25: suggest splitting into two sentences and making the second sentence a more concrete statement of the reactants/conditions necessary for chemodenitrification, similar to that of L84-87. Not all readers readily familiar with chemodenitrification. | The sentence was split accordingly. The parameters upon which chemodenitrification is dependent are now defined more clearly.

[Figure]

L34: Consider adding, "of NO2- consumption" to the sentence, "dominant process..." | It was added.

L34-35: Consider defining abiotic N2O production as one endpoint of chemodenitrification. | We added a defining statement.

L50-52: Example of use of 'non-enzymatic' and 'abiotic'. Please define and then be consistent with use of non-enzymatic vs abiotic. Suggest using just one term, abiotic is more common, I believe. | For clarity "Non-enzymatic" was removed and "abiotic" was used throughout.

L56-58: Check equations. Are H2O and H+ flipped? L60: Cu2+? Should this be Fe2+? | Thank you for catching the error on citing Eq 3 as with Cu when Fe was listed in Equation. We revised in depth the reports on Eq 3 (with Fe) or the possible alternative with Cu, and concluded that the evidence provided to support the feasibility of Fe based equation was limited, the conditions required for its occurrence in nature very unlikely and could cause some unintended confusion. As per Cu, it will not have the role as reductant in reaction either. Hence we removed equation 3, indicated the lack of knowledge on the potential abiotic reduction of N2O and the unlikelihood that this reaction catalyzed by Cu in peat soils. Now L 60-63. We believe this address in depth the point of reviewer.

L66-L72: Found this section a bit confusing. A sentence or two introducing and/or contextualizing these reactions would help readers who are not familiar with these processes. Suggest adding an explanation of how these processes could be affected by sterilization techniques, if indeed that is the point of this part of intro. . . | A sentence was added to clarify that organic functional groups are possible reactants to NOx– just like soil metals (now L67-68). Also, a sentence in L81-82 was extended to reflect that those pools are affected by the methods or techniques used for sterilization: "but affecting metals, organic matter or other pools".

L73-74: Equations absent | Eq 4 (we changed it to Reaction Scheme 1) is a scheme

showing nitrite reaction with phenolic groups that perhaps pdf version did not show in reviewers copy. We have added it, we are making sure the correct technical name (Reaction Scheme) is used and we are checking that it appears in file. Now L76

L124: Were all treatments prepared in an anaerobic glove box prior to incubation? | Indeed, a clarifying sentence was added in L140 in initial MS and now L143-144.

L175: N2 or H2 glove box? Not sure if it would matter, just checking for consistency | This was corrected and specified as 0.5% H2 in N2. Now L 180-181

L268: Zn data is shown in Fig. 3, referring to another figure or a typo? | There is a small confusion here Zn data mentioned in line is in reference to changes in native Zn metal concentration in samples, while Figure 3 has a legend indicating Zn as the treatment (ZnCl2). As for correction, we included in the sentence a term of "metals in soil samples" (L 277); and all figure legends have received the inclusion of a sentence explaining that "X-axis represents treatments where Zn=ZnCl2, etc." Thank you for pointing this out, figures will be clearer with this info.

L292: increased the FI relative to what? Suggest reminding readers of what the baseline and controls were for Table 1, or maybe I missed this explanation earlier. Not sure if referring to a change in time or change relative to live or relative to +/- nitrite etc. | We added a clarifying statement indicating that the first value is from live soil baseline prior to NO2– incubation in now L296.

L307: where is r2 value coming from, was there a regression analysis done? | We added a clarifying statement in now L311-312. A linear regression over full experimental data range was conducted.

L327: However, to me many of the NO2- consumption lines do not look highly linear in first 48 hrs (Fig. 5). | That is correct, we stated now that this linearity is more reflected in the N2O curve. Now L 337

L380: none -> no | Corrected.

L392: delete 'accompanied by'? or check grammar of this sentence | Corrected.

Please also note the supplement to this comment:
https://www.biogeosciences-discuss.net/bg-2019-282/bg-2019-282-AC1-
supplement.pdf

---

## Author Comment (AC2) · 4 Oct 2019

Dear reviewer

Thanks so much for the kind words and appreciation for what is aimed to be contributed here. Positive feedback is always key and the comments offered by your assessment reflect the effort we put on this contribution.

below find the point by point answer to some specific points raised that require some addressing.

L 60: explain that Cu is required for nitrous oxide reductase. | We revised the inclusion

on the note of Cu in this section of the MS and clarify that its concentrations are unlikely to abiotically promote N2O reduction reaction. Although we acknowledge that Cu is required for nitrous oxide reductase we deemed that the mentioning of it did not change the point we were describing and decided not to include to avoid confusion indicating the need of Cu in a biotic process when describing abiotic processes. Now L 60-63.

L 73: Equation 4 is really more of a Figure, no? | That is correct, since we equated the number of elemental atoms albeit in different structures, we chose the "reaction scheme" term as the most appropriate one to describe this item. we changed the name to Reaction Scheme 1 now.

L155: Explain what a methanizer is. | We included an explanation in now L159: "For CO2 measurements, a methanizer (which reduces CO2 to the detectable CH4 via a Ni catalyst at 355°C) was run in line before the FID.

It may be pertinent to mention that decomposition of nitrous acid (HNO2) could be occurring at the especially low pH conditions of the HgCl2 addition. Might be useful to consider whether you think this could be contributing to any of the dynamics observed in this treatment, where pH was 3.6. See Park and Lee, 1988 (J. Phys. Chem. v92. p6294). | Thank you, we now made aware of that potential contribution in L 370-371.

L380: . . . since no enzymes. . . | Corrected.

L392: . . . accompanied by ICP-OES. . . | Corrected.

Please also note the supplement to this comment:
https://www.biogeosciences-discuss.net/bg-2019-282/bg-2019-282-AC2-supplement.pdf

―――――――――――――――――――

**Supplement:**

[revised manuscript text omitted]